# Conformational Changes and Unfolding of β-Amyloid Substrates in the Active Site of γ-Secretase

**DOI:** 10.3390/ijms25052564

**Published:** 2024-02-22

**Authors:** Jakub Jakowiecki, Urszula Orzeł, Przemysław Miszta, Krzysztof Młynarczyk, Sławomir Filipek

**Affiliations:** Faculty of Chemistry, Biological and Chemical Research Centre, University of Warsaw, 02-093 Warsaw, Poland; jjakowiecki@chem.uw.edu.pl (J.J.); u.orzel@student.uw.edu.pl (U.O.); pmiszta@chem.uw.edu.pl (P.M.); kmlynarczyk@chem.uw.edu.pl (K.M.)

**Keywords:** Alzheimer’s disease, gamma-secretase, membrane proteolysis, beta-amyloid, substrate trimming

## Abstract

Alzheimer’s disease (AD) is the leading cause of dementia and is characterized by a presence of amyloid plaques, composed mostly of the amyloid-β (Aβ) peptides, in the brains of AD patients. The peptides are generated from the amyloid precursor protein (APP), which undergoes a sequence of cleavages, referred as trimming, performed by γ-secretase. Here, we investigated conformational changes in a series of β-amyloid substrates (from less and more amyloidogenic pathways) in the active site of presenilin-1, the catalytic subunit of γ-secretase. The substrates are trimmed every three residues, finally leading to Aβ_40_ and Aβ_42_, which are the major components of amyloid plaques. To study conformational changes, we employed all-atom molecular dynamics simulations, while for unfolding, we used steered molecular dynamics simulations in an implicit membrane-water environment to accelerate changes. We have found substantial differences in the flexibility of extended C-terminal parts between more and less amyloidogenic pathway substrates. We also propose that the positively charged residues of presenilin-1 may facilitate the stretching and unfolding of substrates. The calculated forces and work/energy of pulling were exceptionally high for Aβ_40_, indicating why trimming of this substrate is so infrequent.

## 1. Introduction

The major histopathological marker of Alzheimer’s disease (AD) is a presence of amyloid plaques composed mostly of the amyloid-β (Aβ) peptides [1,2,3]. The majority of AD cases are late-onset, after the age of 65, and about a third of the people with late-onset Alzheimer’s are 80 and older. A clear cause of AD is still unknown and AD is the only disease among the top 10 killers that has no effective treatments. There are likely different multifactorial contributors including age, environment, biology, and genetics, which can increase the risk for this disease. The evidence suggests that the disease process begins up to 20 years before the first cognitive symptoms are observed [4,5]. It is possible that the drug treatments may only be effective in the early stages of AD and not when the disease has already caused extensive neurodegeneration. Therefore, knowledge of how the substrates behave in the active site of γ-secretase (GS) complex, which generated the Aβ peptides, is so important for understanding the mechanisms of the substrate cleavage. This can lead to designing orthosteric or allosteric small molecules to block or modulate the enzyme function [6] and ultimately designing the substrate-selective inhibitors. GS modulation could also be beneficial in the treatment of other diseases, including cancer drug resistance, as the GS/Notch pathway promotes transformation of cancer cells into cancer stem cells [7].

The GS substrates generally require shedding of their extracellular domains prior to intramembrane proteolysis. Shedding of amyloid precursor protein (APP) by α-secretase generates APP-C83 [8] (on the nonamyloidogenic pathway), while shedding by β-secretase generates APP-C99 (on the amyloidogenic pathway), which are next cleaved by GS [9,10]. APP-C83 cleavage products are non-toxic and soluble, and they have names starting with the letter “s”, e.g., sAβ_43_, while APP-C99 cleavage products are toxic and can form insoluble oligomers. They are longer by 16 residues compared to their counterparts from the nonamyloidogenic pathway (Figure 1). Successive cleavage of APP-C99 and then trimming it by GS leads to the release of the Aβ peptides, including the most abundant Aβ_40_, while the most toxic is Aβ_42_ [11,12]. Aβ_40_ and Aβ_42_ are produced in a 10:1 ratio. Shorter products, namely Aβ_37_ and Aβ_38_, can also be produced but they are detected in very low amounts [13]. Conformational changes in APP are extremely important for its cleavage as was recently shown for APP FAD mutation T714I, which severely reduced the cleavage, but the secondary APP mutations restore the cleavage of APP T714I [14]. Several other substrates were also analyzed as to how they interact with individual transmembrane helices of GS [15]. On the other hand, the conformational changes in distant areas of GS can also influence the substrate cleavage, as was shown for mutations in APH-1 [16] and in case of microbial infection (HSV-1 membrane glycoprotein US7) that promotes AD via interacting with and modulating GS [17].

The recent cryo-EM structure of GS with APP-C83 substrate, i.e., after α-secretase cleavage and before the first cut by γ-secretase, was solved with a very good resolution of 2.6 Å (PDB id: 6IYC) [18], allowing us to see atomic details of the structure. The GS complex consists of the catalytic subunit presenilin-1 (PS-1), which is associated in a 1:1:1:1 stoichiometry with three membrane proteins: PEN-2, APH-1, and NCT (Figure 2a). PEN-2 (presenilin enhancer 2) is composed of one entire transmembrane helix and two half-helices, with a turn between them in the middle of the membrane. APH-1 (anterior pharynx defective 1) is composed of seven transmembrane helices with relatively short loops between them, while NCT (nicastrin) has only one transmembrane helix and a very large ectodomain, which plays a role in substrate recognition. PS-1 contains nine transmembrane helices: TM1–TM6, forming the N-terminal part, and TM7–TM9, forming the C-terminal part. The long loop between helices TM6 and TM7 is cleaved during maturation of the GS complex, but both parts of PS-1 stay together. They also participate in the substrate cuts together since the catalytic residues, D257 and D385, are located in adjacent helices TM6 and TM7. Part of the loop between these helices forms a β-sheet with a substrate (Figure 2b).

The mechanism of substrate processing by GS was the subject of many investigations, including theoretical calculations. Using the molecular dynamics (MD) simulations, Hitzenberger and Zacharias [19] identified the most probable enzyme–substrate interfaces for APP-C99, Aβ_49_, Aβ_46_, and Aβ_43_ in the binding site of GS; however, they used an earlier structure of GS (PDB id:5FN3) [20] where no β-sheet is visible between GS and its substrates. They also performed in silico alanine scanning for all residues of all analyzed substrates and found that most substitutions to alanine destabilized the complex. However, the mutations of residues in the juxtamembrane region, S26A, N27A, and K28A, increased the complex’s stability. It is known from the experimental studies [21] that mutations of N27 and K28 of APP-C99 to apolar residues lead to much shorter products than Aβ_40_, indicating that trimming continued. N27 and K28 could anchor Aβ peptides to the headgroup region of the lipid bilayer, preventing excessive trimming of the substrate.

Bhattarai et al. [22] performed Gaussian accelerated molecular dynamics of GS with Notch, APP, and mutated APP. The POPC model membrane without cholesterol was used for all simulations and they lasted from 300 ns to 2 μs. Three mutations of APP were studied for their effect on cleavage preferences and it was found that the M51F mutation shifted the ε cleavage site of APP by one residue to the amide bond between Thr48 and Leu49. In their subsequent publication [23], they investigated the mechanism of tripeptide trimming of Aβ_49_ by GS using MD simulations as well as biochemical methods on Aβ_49_ and its mutants. They used the Pep-GaMD method to accelerate structural rearrangements of both the enzyme and substrate.

In our previous studies [10], we investigated the role of cholesterol in the recognition and binding of amyloidogenic fragments of APP (substrates Aβ_43_ and Aβ_45_, leading in the next cut to less amyloidogenic Aβ_40_ and more amyloidogenic Aβ_42_, respectively). Although the substrates differ by only two residues at the C-terminus, we have found that their interactions with the binding site of GS were statistically different, and the differences were the largest (excluding the unfolded part of the N-terminus) at the N-terminal helical part of the substrate that sticks out from PS-1 and has contact with a membrane. It was also found that at 50% cholesterol levels, an average of two cholesterol molecules interacted with the substrate helix, possibly directing it towards the catalytic site and stabilizing it.

High cholesterol levels were found experimentally in human brains, and were used in theoretical investigations. Chan et al. [24] performed Lipidomic Analysis of mouse and human brain with Alzheimer’s disease. They found that, depending on the brain region, 44–48% of the lipidome measured in the analysis was made up of free cholesterol. It was also reported [25] that approximately 70% of the cholesterol in the adult brain is located within myelin membranes, where it plays a pivotal role in insulating axons. High cholesterol levels, up to 60%, were also used in theoretical studies [26] on the influence of membrane lipid composition on the structure and activity of GS.

In the current study, we investigated conformational changes in the C-terminus of Aβ substrates in the active site of GS to see differences between less amyloidogenic pathway substrates, Aβ_46_ and Aβ_43_, and more amyloidogenic pathway substrates, Aβ_48_ and Aβ_45_. The extended C-terminus of substrates is only three residues long, since the subsequent cuts are performed every three residues, so it can form a short β-sheet or only a β-bridge (Figure 2c). The testing of the stability of such a shortened β-sheet of the substrate as well as of differences in their flexibility was performed using all-atom molecular dynamics (MD) simulations. Moreover, using steered molecular dynamics (SMD) simulations to induce unfolding of the C-terminus of the substrate, we studied conformational changes in the whole membranous part of the substrates. For this purpose, we employed the whole set of less amyloidogenic pathway substrates (Aβ_49_, Aβ_46_, Aβ_43_, and Aβ_40_) to see how different positions of the same residues of these substrates influence the unfolding process, and the rationale behind terminating the trimming process on a specific short substrate.

## 2. Results and Discussion

To investigate the stability of four intermediate products of GS in the active site of this enzyme, we performed eight 500 ns all-atom MD simulations of the whole GS complex using two repeats for each case: Aβ_48_ and Aβ_45_ (from more amyloidogenic path leading to Aβ_42_) and Aβ_46_ and Aβ_43_ (from less amyloidogenic path leading to Aβ_40_). GS trims the major intermediate products by mostly three, and sometimes four, residues [3,27]; however, the mechanism of the trimming is still unclear. The recent cryo-EM structures with substrate APP-C83 [18] and with human Notch fragment Notch-100 [28] provided nearly the same conformation of the main chain of both substrates in the active site of GS. This may indicate the same or a similar mechanism for the first cleavage of the substrate and then trimming it. Yang et al. proposed [28] for cleaving APP that the β-sheet between the substrate and the TM6-TM7 loop could still exist for shorter substrates and stabilize their correct conformation for the next cut. The β-sheet for APP-C83 is five residues long (Figure 2b), while for the shorter substrates, only three residues long (Figure 2c), since the C-terminal part is truncated. Because of this difference, the question of the stability of such complexes arises.

We first focused on the area containing catalytic residues of GS and a scissile bond of a substrate. In Figure 3, one can see correlations of two distances: between the catalytic residues, and between the closer catalytic residue (Asp385) to the peptide bond between (n − 4) and (n − 3) substrate residues, i.e., the bond that will be cleaved. For calculation of the above distances, the C_γ_ (CG) atoms of aspartate residues were used to exclude rotations of carboxyl groups, and the carbonyl oxygen (atom name O) of the substrate residue (n − 4).

In case of the Aβ_46_ substrate (Figure 3a), the catalytic residues remain stable with a distance between them in the range of 6.5–7.5 Å, while the catalytic residue Asp385 forms a stable hydrogen bond to the scissile bond. For the Aβ_43_ substrate (Figure 3b), the distance between catalytic residues is shorter, about 6.5 Å, but the distance to a scissile bond is longer than for the previous substrate, in the range of 4.0–6.0 Å.

For the longest substrate, Aβ_48_, (Figure 3c) there are two stable states: (i) characterized by a long distance between the catalytic residues and a direct contact (a hydrogen bond) of Asp385 to a scissile bond; (ii) characterized by a shorter distance between the catalytic residues and the lack of a direct contact (a hydrogen bond via water molecule) of Asp385 to a scissile bond. For the Aβ_45_ substrate (Figure 3d), there is a hydrogen bond of Asp385 to a scissile bond nearly all the time; however, the distance between catalytic residues is larger (in the range of 7.0–9.0 Å). Comparing plots for less amyloidogenic pathway substrates (Aβ_46_ and Aβ_43_) with more amyloidogenic pathway substrates (Aβ_48_ and Aβ_45_), one can see that shorter distances between catalytic residues are associated with both Aβ_46_ and Aβ_43_. Since the production of Aβ_40_ to Aβ_42_ is 10:1, this means that holding the two catalytic residues together is more important for cleavage than maintaining a short and constant distance to the scissile bond.

We then studied the stability of the extended parts of the substrate and PS-1 by observing the distances of last and last but one residues of substrate to nearby residues of the TM6-TM7 loop, forming an antiparallel PS-1 β-sheet (β2 in Figure 2c). For any substrate, we observed close proximity of the last residue (n) of the substrate and the Lys380 residue of PS-1, and between the last but one residue (n − 1) of substrate and the Leu381 residue of PS-1, marked with a red dashed square in Figure 4. The patterns of preferable distances are nearly the same for Aβ_43_ and Aβ_45_ (Figure 4b,d); however, they are much different for Aβ_46_ and Aβ_48_ (Figure 4a,c). There is only one minimum for Aβ_48_, indicating the high stability of the C-terminus of this substrate. The distance of the Aβ_48_ scissile bond to the catalytic residue Asp385 is short, but the distance between catalytic residues is large (Figure 3c), compared to that for Aβ_46_ (Figure 3a).

On the other hand, we did not observe any close proximity between the (n − 1) residue of the substrate and the L432 residue located in the TM8-TM9 loop of PS-1 (β4 in Figure 2c) and forming a β-bridge (a single pair of β-sheet hydrogen bonds) present in the 6IYC structure. It seems that this β-bridge is not stable for substrates after the initial cleavage (ε-cleavage).

More details on how the secondary structure of the substrates changes over time can be seen in Figure 5. The formation of a stable and regular β-sheet, as is seen in the 6IYC structure, was not possible; however, the β-bridge was formed especially for the last but one residue of the substrate. The most stable β-bridge was seen for the Aβ_48_ substrate, where it was visible in both MD simulations (Figure 5c)—39.9% on average. For the Aβ_43_ substrate, the β-bridge was only seen in one simulation but it was quite stable (Figure 5b)—38.7% on average. For other substrates, the β-bridges (or β-sheets) were also formed but they were less stable—24.9% for Aβ_45_ (Figure 5d) and 9.4% for Aβ_46_ (Figure 5a), on average. The timelines correlate well with the distances between the C-terminal part of the substrate and a β-sheet formed by the TM6-TM7 loop (Figure 4), confirming that the longest substrate, Aβ_48_, was the most stable, while Aβ_46_ was the least stable, and that the interactions with PS-1 loop were indeed stabilizing the extended conformation of the C-terminus of the substrate.

The C-terminus of the trimmed substrate can be pulled for next cut by the cleaved fragment, AICD (APP intracellular fragment), as was shown in [23], or three-residue fragments from trimming, but such pulling can be augmented by positively charged residues, including Lys380, located just behind the active site, which was suggested by Koch et al. [13] based on all-atom classical 200 ns MD simulations. In our simulations, we have also seen interactions of the positively charged amine group of Lys380 and the negatively charged C-terminus of the substrate (Figure 6). This is especially visible for the shortest substrates: Aβ_43_ (Figure 6a) and Aβ_45_ (Figure 6b). For longer substrate Aβ_46_ (Figure 6c), the salt bridge to Lys380 is less frequent and it nearly disappears for Aβ_48_ (Figure 6d). Figure 6e shows such a binding event with Lys380, but other positively charged residues, Arg269 and Arg377, are also able to form similar ionic interactions and replace Lys380.

In some MD simulations, especially for the longest substrate Aβ_48_, we observed the formation of a salt bridge between the charged carboxyl group of the C-terminus of the substrate and residues Arg269 (Figure 7a) and Arg377 (Figure 7b) of PS-1. Both arginine residues are located in the TM6-TM7 loop. These residues are flanking the channel for removal of the trimmed products, the tripeptides. The formation of these salt bridges correlates with the stability of the last residues of the substrate, since Aβ_48_ forms the most stable extended C-terminal parts (Figure 4c and Figure 5c). The formation of a salt bridge of the substrate C-terminus with Lys380 (Figure 6e) as well as Arg269 and Arg377 of PS-1 is exactly at a distance of three residues of substrate to the scissile bond, which is located close to the catalytic residues (Figure 7c). Strong interactions with the positively charged residues possibly prevents further unfolding of the substrate and facilitates trimming, mostly by three residues. The residues Arg377 and Lys380 are located in a flexible loop so they can facilitate the removal of trimmed tripeptides from the active site of GS, and then support stretching of the substrate to prepare it for the next cut. The role of Arg377, apart from Lys380, in contributing to the substrate unfolding was also proposed by Koch et al. [13]. The salt bridges to both arginine residues are not stable, and the same is visible for Lys380 (Figure 6d), so it is probable that the mechanism of pulling of the longest substrates by positively charged residues is replaced by pulling of positively charged N-terminus of AICD, which was precisely shown by Bhattarai et al. [23] for the Aβ_49_ substrate.

Both arginine residues of PS-1, Arg269 and Arg377, are important for APP cleavage, since mutations of them were found in familial AD patients’ brains: R269H and R377M [29]. Other AD-derived mutations were investigated by Sun et al. [30], in a series of PS-1 mutations, for their impact on GS activity. They found that the R269G mutant significantly lowered amounts of produced Aβ_40_ and Aβ_42_, and increased the Aβ_42_/Aβ_40_ ratio. Another mutant, R377W, nearly stopped producing Aβ_40_ and Aβ_42_. These experimental data are in line with the hypothesis of the contribution of Arg269 and Arg377 to the substrate unfolding, but more experiments should be done to understand the precise roles of these residues in the processing of the APP substrate. The biological effects of mutations of residues Arg269, Arg377, and Lys380 are summarized in Table 1.

In order to study unfolding events of substrates until reaching the conformation corresponding to the next cleavage event, we performed a series of SMD simulations using the GS-SMD web server [35]. We considered the following cases: (1) unfolding of the C-terminal fragment of Aβ_49_ by three residues until the conformation ready for next trimming to Aβ_46_ was reached (Figure 8a); (2) unfolding Aβ_46_ → Aβ_43_ (Figure 8b); (3) unfolding Aβ_4_*_3_* → Aβ_40_ (Figure 8c); (4) unfolding Aβ_40_ → Aβ_37_ (Figure 8d). Although the initial conformation of the substrate before pulling is the same because of employing the threading technique, the final conformation after pulling by three residues is much different. For Aβ_49_ and Aβ_46_, the substrate retains a helical form, whereas for Aβ_43_ the helix is completely unfolded (Figure 8c). The reason for such behavior is the presence of polar and charged residues at the extracellular edge of the membrane. They keep staying behind this edge. However, when the pulling force is larger, they enter the membrane but form a small helix to minimize contact with the hydrophobic core of the membrane (Figure 8d). Plots of force and work/energy for each SMD simulation are shown in Appendix A. Timeline plots for these SMD simulations are presented in Appendix A. In each figure, the frame with the closest distance of scissile bond to the catalytic residues (trimming conditions) is indicated.

Upon the unfolding and trimming of the Aβ substrate, its N-terminal polar residues are still getting closer to the membrane environment. Polar residues 26–28 of Aβ, forming the SNK motif, remain on the extracellular site in the first (Aβ_49_ → Aβ_46_), second (Aβ_46_ → Aβ_43_), and third unfolding (Aβ_43_ → Aβ_40_). However, during the 4th unfolding, Aβ_40_ → Aβ_37_, the SNK motif enters the membrane, which is correlated with a rise in the SMD pulling force (Appendix A). This means that in order to trim Aβ_40_ by the following three residues, the system would need to overcome a high energetic barrier associated with pulling the polar residues into the membrane hydrophobic environment. Therefore, the polar Aβ residues form an anchor that stops the trimming on the Aβ_40_ product. Our findings correspond with the results presented by Koch et al. [13], highlighting the role of the extracellular domain of the APP substrate in delimiting the Aβ product length. The experimental results showed that any single mutation in positions Aβ26–28 (SNK residues) for apolar residues (alanine or phenylalanine) significantly shifts trimming towards Aβ_37_ production.

In SMD simulations of the first cleavage, Aβ_49_ → Aβ_46_, the mean work required to pull the substrate by three residues is 38.4 kJ/mol (Appendix A); in the second cleavage Aβ_46_ → Aβ_43_, the mean work equals 47.4 kJ/mol (Appendix A); in the third cleavage Aβ_43_ → Aβ_40_, it is 86.7 kJ/mol (Appendix A); and in the 4th, rarely occurring cleavage, Aβ_40_ → Aβ_37_, 147.4 kJ/mol (Appendix A). Therefore, we observe a progressive increase in the energy barrier in the succeeding cleavages. Tokens to individual SMD simulations deposited on the GS-SMD server are presented in Appendix A. The box-plot showing the calculated work needed to unfold the substrate by three residues is presented in Figure 9. Simulations of the first and second cleavage have comparable work/energy values. On the other hand, the difference of work between the third and first cleavage equals 48.3 kJ/mol and has a *p*-value of 0.008 (<0.05, statistically significant); the third and second cleavage have a difference of 39.3 kJ/mol and a *p*-value of 0.014 (<0.05, statistically significant). In case of a difference between the 4th and first (rarely occurring) cleavage, the work difference is 109.0 kJ/mol with a *p*-value of 0.0001 (<0.05, statistically significant); between the 4th and second the work difference is 100.0 kJ/mol, *p*-value 0.0002 (<0.05, statistically significant); for the third and 4th cleavage, the work difference equals to 60.7 kJ/mol, *p*-value 0.012 (<0.05, statistically significant). The increasing work we see represents a higher energy barrier that needs to be overcome for the next cleavage to happen and therefore explains why the fourth cleavage, trimming Aβ_40_ to Aβ_37_, is rarely observed.

## 3. Methods and Materials

### 3.1. Preparation of the Structure

For MD simulations, we employed cryo-EM structure of γ-secretase (GS) with APP-C83 substrate (PDB id:6IYC) [18]. The N-terminus of substrate is not visible except for residues 1–6 interacting with NCT. The missing five residues 7–11, which are unfolded, were reconstructed. The hydrogen atoms were added in YASARA, at a pH of 7.4 and with the optimization of the hydrogen bond network. The counterions Na^+^ and Cl^−^ were added to make the system neutral, and we set the salt concentration to a physiological level of 0.15 M. The mutation of one of the catalytic residues, D385A, introduced to the cryo-EM structure to prevent substrate cleavage by GS, was reversed. Then, the restored residue, D385, was protonated to create the proper catalytic environment for this aspartic protease enzyme. Such protonation is in agreement with a recently published pH replica exchange molecular dynamics (pH-REMD) simulation of GS [36]. The POPC (1-palmitoyl-2-oleoyl-*sn*-glycero-3-phosphocholine) membrane with 50% cholesterol concentration was prepared in CHARMM-GUI [37]. The long loop between TM6 and TM7 helices (residues 292–375) of PS-1 was not reconstructed to not introduce additional parts, which can influence the interactions of β-sheet with the C-terminal part of the substrate. The existing in 6IYC structure β-sheet, linking visible parts of TM6-TM7 loop and the APP-C83 substrate, was stable in MD simulation, even in the absence of the rest of this loop. The placement of studied substrates, Aβ_43_, Aβ_45_, Aβ_46_, and Aβ_48_, to the binding site of PS-1 was done using the threading procedure with the sequence of a given substrate applied to the 6IYC structure in such a way to mimic the next cleavage event. The C-terminal fragment of the substrate was extended by three residues behind the cleavage site of PS-1 (Figure 2c), i.e., in case of Aβ_43_, the residue no. 40 was placed next to the catalytic residues, and for Aβ_45_, it was the residue no. 42, etc.

### 3.2. Molecular Dynamics (MD) Simulations

We performed all-atom MD simulations using the AMBER 18 program [38] with a standard all-atom force field CHARMM36 [39]. The TIP3P water model was employed, which was parametrized to use with CHARMM force fields. The whole modeled system contained about 260,000 atoms, including about 250 lipid molecules (cholesterol and POPC). The 50% cholesterol corresponds to the raft-like membranes where GS is the most effective. The average dimensions of the periodic cell were 130 Å × 130 Å × 170 Å. The whole system in the membrane was subjected to a restrained energy minimization (5000 cycles). The first 2500 minimization cycles were performed with a steepest-descent method, and after that, the conjugate gradient was employed in the AMBER program. Next, a six-step equilibration was performed (375 ps in total) at a constant pressure and temperature (NPT ensemble; 310 K, 1 bar). During the equilibration, the restraints were released gradually until the last step (which lasted 100 ps), in which no restraints were used. In the production simulations, as well as in the last three equilibration steps, all bond lengths to hydrogen atoms were constrained using the SHAKE [40] algorithm, allowing us to use a longer time step of 2 fs instead of 1 fs. Van der Waals and short-range electrostatic interactions with a cutoff of 12 Å and a 10–12 Å F-switch function were used. Long-range electrostatic interactions were computed using the Particle Mesh Ewald [41] summation scheme.

### 3.3. Steered Molecular Dynamics (SMD) Simulations

We performed a series of simulations of GS with Aβ peptides in our server GS-SMD [35]. The server combines an implicit membrane methodology, IMM1 [42] with SMD, and allows the observation of the process of unfolding a chosen GS substrate by the pulling of any selected atom in any direction with a constant velocity. We selected the C-terminus of the APP fragment substrate and performed pulling in a default direction. This direction is based on the expected final position of the substrate C-terminus that forms a β-sheet with PS-1 as it is observed in cryo-EM structure PDB id:6IYC. We considered the substrates Aβ_49_, Aβ_46_, Aβ_43_, and Aβ_40_, investigating their unfolding by 3 residues each, that represent the following cleavage events: Aβ_49_ → Aβ_46_, Aβ_46_ → Aβ_43_, Aβ_43_ → Aβ_40_, and Aβ_40_ → Aβ_37_. To obtain statistically valid data, we performed 8 repetitions for each substrate. We used default settings: the simulation length was 15 ns, coordinates were saved in 50 frames, and the pulling speed was 1.0 Å/ns. The work required for pulling in each conducted simulation was calculated for a frame characterized with the closest distances between CG atoms (center of carboxylic group) of catalytic residues D257, D385, and the O atom of the n − 3 (third from last) substrate residue, as shown in Figure 10. We rejected those repetitions with the shortest distance between catalytic residues larger than 8 Å and with a sum of distances between the n-3 substrate residue and catalytic residues longer than 11 Å. Since we used an implicit membrane-water methodology, no explicit cholesterol molecules were used. The membrane thickness, i.e., the hydrophobic core, was set to 31 Å (default value in GS-SMD).

We included trimming Aβ_40_ → Aβ_37_ to investigate why the Aβ processing is typically hampered at this point, leaving the Aβ_40_ peptide as the main product. The analysis of SMD frames characterized with the shortest distance between the n-3 substrate residue and the catalytic residues showed that the mean value of the sum of these distances equals 9.7 Å, on average, and the mean distance between the catalytic residues is about 6.5 Å. Therefore, the proximity of the abovementioned residues allows the catalytic cleavage to occur. The typical final distances in each repetition are presented in Figure 10.

The figures presented in this paper were prepared in VMD v.1.9.4 [43], YASARA 22.05.22 [44], PyMOL 2.5.0 [45], and Matplotlib library 3.7.3 [46] in Python 3.8.

### 3.4. Statistical Procedure and the Box-Plot

We compared the average work/energy needed to pull the substrates of different length (i.e., Aβ_49_, Aβ_46_, Aβ_43_, Aβ_40_) towards the conformation required for the next cleavage to occur. To test whether the differences in work/energy are statistically valid, we performed one-tailed two-sample Student’s *t*-test (independent *t*-test) using the Social Science Statistic service (https://www.socscistatistics.com/tests/studentttest/default.aspx (accessed on 13 November 2023)). A two-sample *t*-test is used to test the null hypothesis that the difference between the means calculated for two populations of values is not statistically significant:$$t=\frac{{\bar{X}}_{1}-{\bar{X}}_{2}}{\sqrt{\frac{\left(n_{1}-1\right)s_{1}^{2}+\left(n_{2}-1\right)s_{2}^{2}}{n_{1}+n_{2}-2}\left(\frac{1}{n_{1}}\right.+\left.\frac{1}{n_{2}}\right)}}$$
where

${\bar{X}}_{1}$ and ${\bar{X}}_{2}$ are the means of the two groups of values.

$s_{1}^{2}$ and $s_{2}^{2}$ are the variances of the two groups of values.

$n_{1}$ and $n_{2}$ are the sample sizes of the two groups of values.

The calculated *t*-value is a measure of difference between the means of two populations relative to the variation within the populations. The *t*-value is used to calculate the probability that the null hypothesis is true (i.e., a possible difference between the means of two populations is caused by a statistical error, and if an infinite number of measurements were taken for each population, their means would be identical). This probability is known as *p*-value. If the *p*-value is sufficiently small (typically below 0.05), it suggests that the observed difference is unlikely to have occurred by chance alone, and thus provides evidence against the null hypothesis.

The box-plot presented in Figure 9 was prepared in the Matplotlib pyplot library. The box extends from the first quartile (Q1) to the third quartile (Q3) of the data, with the internal line at a median. The whiskers extend from the box to the farthest data point lying within 1.5× the inter-quartile range (IQR) from the box. The presented data do not have any outliers (flier points).

## 4. Conclusions

Using MD simulations, it was possible to study the conformational changes and flexibility of extended C-terminal parts of four amyloid peptides, which are trimmed by GS: Aβ_48_ and Aβ_45_, leading to the more amyloidogenic product Aβ_42_, as well as Aβ_46_ and Aβ_43_, leading to the less amyloidogenic product Aβ_40_. The C-terminus was extended by three residues behind the catalytic residues of PS-1 using the threading procedure with the sequence of a given substrate applied to the 6IYC structure. It was found that keeping two catalytic residues together is more important for the substrate cleavage than keeping a short and steady distance from the catalytic residues to the scissile bond. The larger flexibility of the C-terminal parts of less amyloidogenic pathway substrates (Aβ_46_ and Aβ_43_) does not preclude keeping the catalytic residues together; on the contrary, we rather see that the excessive stabilization of the substrate C-terminus (Aβ_48_) is associated with separation of the catalytic residues. Based on conducted MD simulations, we can also hypothesize that the positively charged residues, Lys380, Arg269, and Arg377, located on the border of the channel for removal of the trimmed products, tripeptides, from the active site of PS-1, may facilitate stretching and unfolding of the C-terminus of the substrate Lys380 for the shortest substrates, and Arg269 and Arg377 for the longer substrates. Moreover, we investigated the process of subsequent trimming Aβ_49_ → Aβ_46_ → Aβ_43_ → Aβ_40_ using the SMD methodology and compared it with a rarely occurring next cleavage Aβ_40_ → Aβ_37_. We observed that acidic and basic residues present in the Aβ sequence are not entering the membrane environment up to the third trimming, resulting in the Aβ_40_ product. The obtained work/energies of pulling indicate that substrate unfolding is hampered by polar residues that do not allow the formation of substrates shorter than Aβ_40_.

Knowing the details of the cleavage mechanism by GS could facilitate drug discovery to control the overproduction of amyloidogenic products Aβ_40_ and Aβ_42_. The experimental studies involving mutations of GS as well as the theoretical approaches using MD simulations lead us to a better understanding of the GS trimming mechanism. This could in the future clarify the substrate specificity of GS in order to design selective inhibitors and modulators.

## Figures and Tables

**Figure 1 ijms-25-02564-f001:**
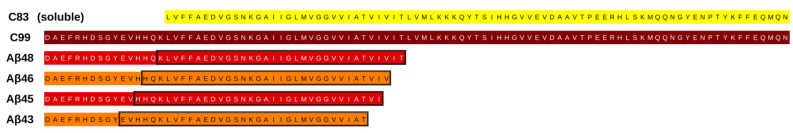
A comparison of cleavage products of amyloid precursor protein (APP). APP-C83 is generated by α-secretase cleavage, while APP-C99 is generated by β-secretase cleavage. These pathways are called nonamyloidogenic and amyloidogenic, respectively. Below are four intermediate products obtained from APP-C99, which are substrates for subsequent cuts by γ-secretase: Aβ_48_ and Aβ_45_ leading to Aβ_42_ (more amyloidogenic product), and Aβ_46_ and Aβ_43_ leading to Aβ_40_ (less amyloidogenic product). Black rectangles denote sequences used to create substrates of the same length for MD simulations.

**Figure 2 ijms-25-02564-f002:**
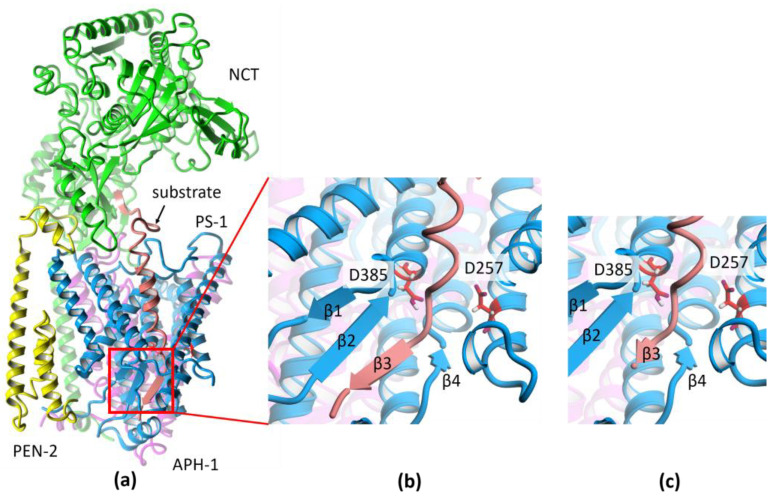
The structure of γ-secretase (GS). (**a**) The cryo-EM structure (PDB id:6IYC) of GS with APP-C83 substrate. Colors of subunits: PS-1 in cyan, APH-1 in purple, NCT in green, PEN-2 in yellow, and the Aβ substrate in salmon. (**b**) Magnification of the catalytic site showing β-sheet formed between a substrate and PS-1. The catalytic residues of PS-1 are shown in red. (**c**) The substrate after threading of substrate sequence into 6IYC structure to obtain the substrate extended by three residues behind the catalytic residues for trimming.

**Figure 3 ijms-25-02564-f003:**
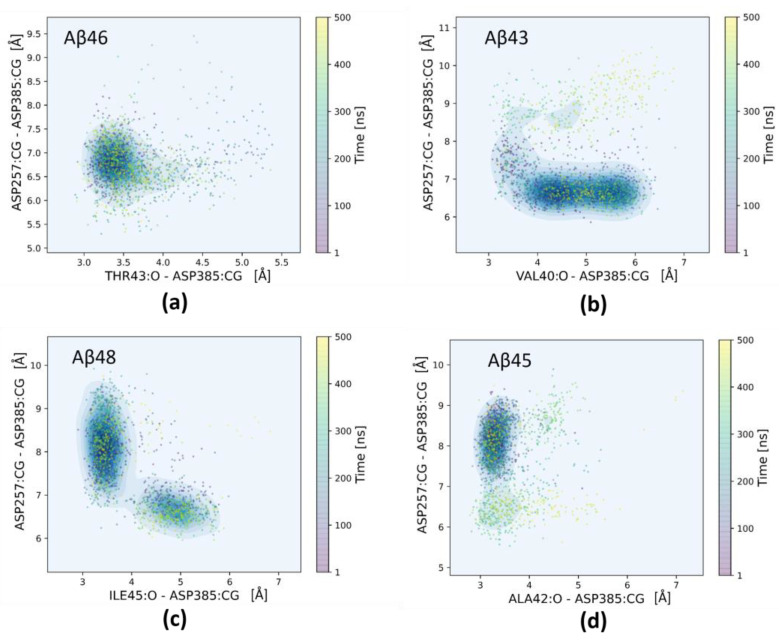
Two-dimensional scatter plots showing distances between the carboxylic group of one of the catalytic residues, Asp385, to the peptide bond of residue that is to be cleaved (horizontal axes) and distances between carboxylic groups of the catalytic residues (vertical axes). Each panel is averaged from two MD simulations. All points are colored according to the MD simulation time from purple (0 ns) to yellow (500 ns). (**a**) Aβ_46_ substrate with a Thr43 scissile bond; (**b**) Aβ_43_ substrate with a Val40 scissile bond; (**c**) Aβ_48_ substrate with an Ile45 scissile bond; (**d**) Aβ_45_ substrate with a Ala42 scissile bond.

**Figure 4 ijms-25-02564-f004:**
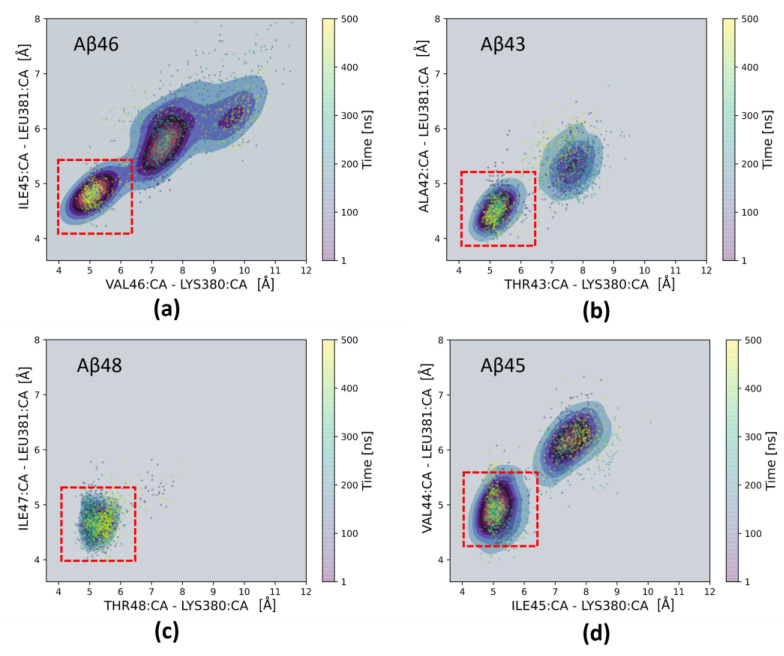
Two-dimensional scatter plots showing distances between C_α_ (CA) atoms of selected residues from the substrate and PS-1: the distance between C_α_ atoms of the last residue (n) of the substrate and Lys380 of PS-1 (horizontal axes), and the distance between C_α_ atoms of last but one residue (n − 1) of the substrate and Leu381 of PS-1 (vertical axes). Each panel is averaged from two MD simulations. Red dashed square indicates the shortest distances between the above residues with a possibility of forming a β-sheet composed of four residues: n and n-1 substrate residues and PS-1 residues 380–381. (**a**) Aβ_46_ substrate; (**b**) Aβ_43_ substrate; (**c**) Aβ_48_ substrate; (**d**) Aβ_45_ substrate.

**Figure 5 ijms-25-02564-f005:**
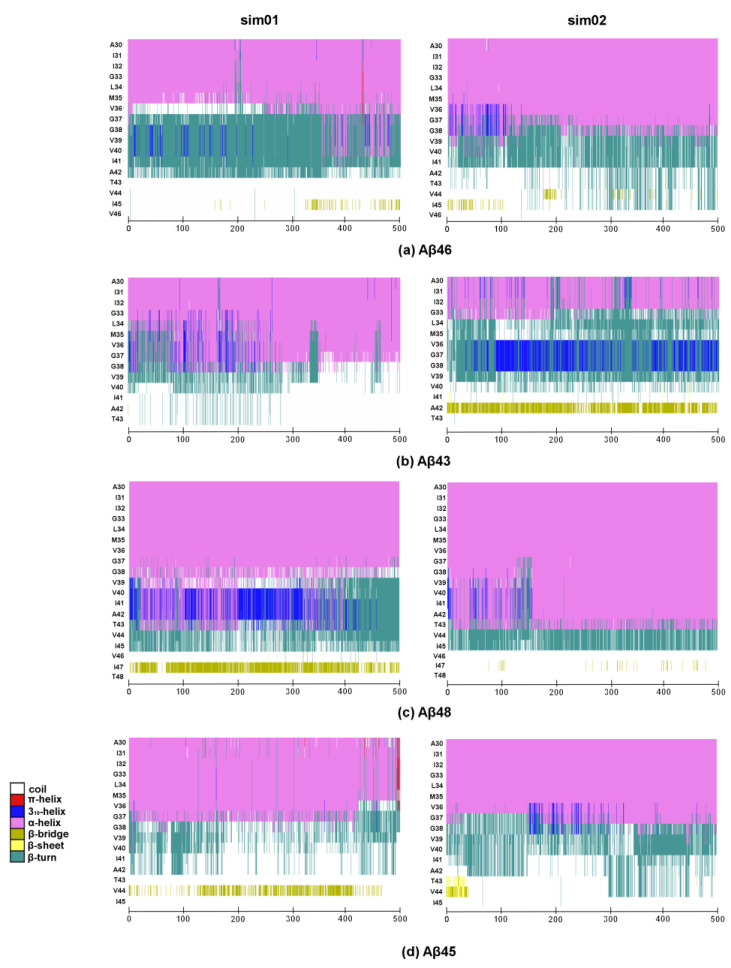
Timeline showing the secondary structure of four substrates during 500 ns of MD simulation. Each simulation is presented in a separate graph over time along the horizontal axis. N-terminal part of substrates is not shown. (**a**) Aβ_46_ substrate; (**b**) Aβ_43_ substrate; (**c**) Aβ_48_ substrate; (**d**) Aβ_45_ substrate.

**Figure 6 ijms-25-02564-f006:**
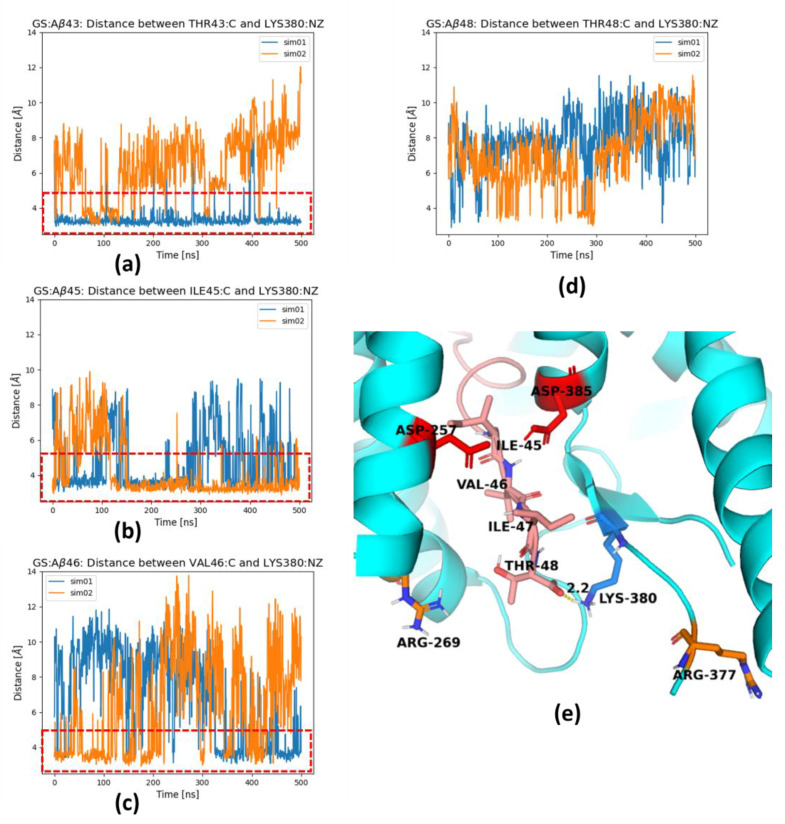
Lys380 residue forms a salt bridge with the substrate C-terminus; (**a**) with Thr43 of Aβ_43_; (**b**) with Ile45 of Aβ_45_; (**c**) with Val46 of Aβ_46_; (**d**) with Thr48 of Aβ_48_; (**e**) 3D structure of Aβ_48_ substrate extended by three residues so the catalytic residues (in red) are localized in proximity of next cut residue Ile45. Red dashed rectangles in panels (**a**–**c**) indicate formation of a salt bridge. Results from two independent simulations in panels (**a**–**d**) are shown in different colors. The hydrogen bonds are indicated by dashed yellow cylinders while distances are shown in [Å]. The catalytic residues are shown in red, PS-1 in cyan and Aβ in salmon.

**Figure 7 ijms-25-02564-f007:**
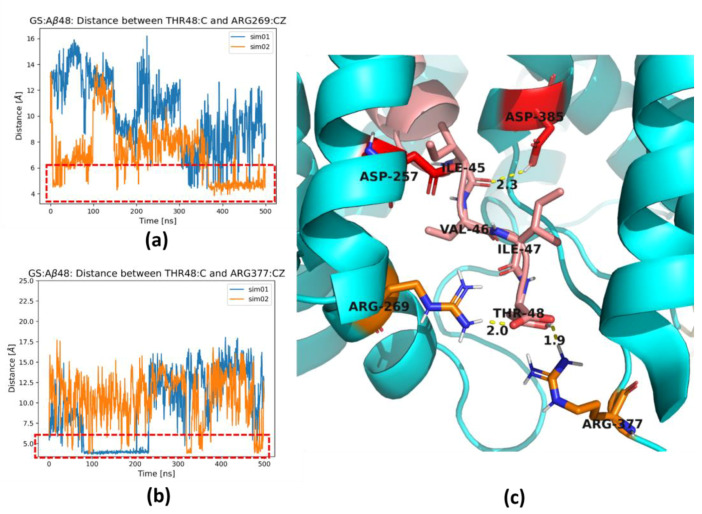
Two arginine residues, Arg269 and Arg377, of PS-1 form a salt bridge with C-terminus of the substrate. (**a**) The distance between the carboxyl terminal group of C-terminus of Aβ_48_ (atom O) and Arg269 of PS-1 (atom CZ). (**b**) The distance between the carboxyl terminal group of C-terminus of Aβ_48_ (atom O) and Arg377 of PS-1 (atom CZ). Results from two simulations are shown in different colors. Red dashed rectangles indicate formation of a salt bridge and a hydrogen bond. (**c**) Magnification of the area showing C-terminal residues of Aβ_48_ substrate (in salmon) and residues of PS-1 (in cyan): two arginine residues, Arg269 and Arg377, in orange and two catalytic residues, Asp257 and Asp385 (in red). The hydrogen bonds are indicated by dashed yellow cylinders while distances are shown in [Å]. PS-1 is shown in cyan and Aβ in salmon.

**Figure 8 ijms-25-02564-f008:**
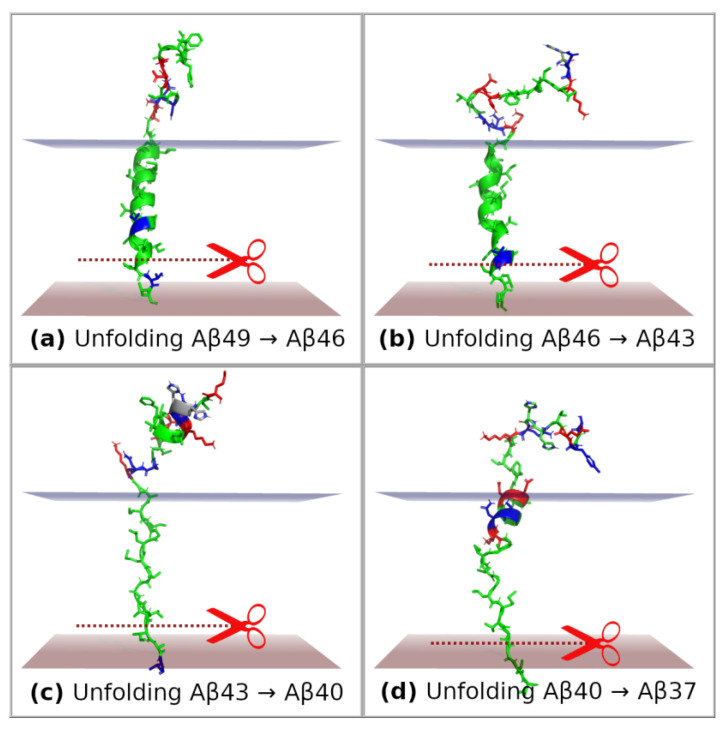
Final representative conformations of substrates after SMD simulations in GS-SMD web server. Unfolding was performed until the conformation ready for next trimming was reached (by three residues). GS is not shown so as not to obscure the substrate. (**a**) Pulling and unfolding of Aβ_49_ to prepare the next cleavage event Aβ_49_ → Aβ_46_; (**b**) unfolding of Aβ_46_ → Aβ_43_; (**c**) unfolding of Aβ_43_ → Aβ_40_; (**d**) unfolding of Aβ_40_ → Aβ_37_. Borders of the hydrophobic core of the membrane are represented by two planes. The polar and charged residues remain on the extracellular side and pull the substrate back. Colors of residues: green—apolar, blue—polar, red—charged.

**Figure 9 ijms-25-02564-f009:**
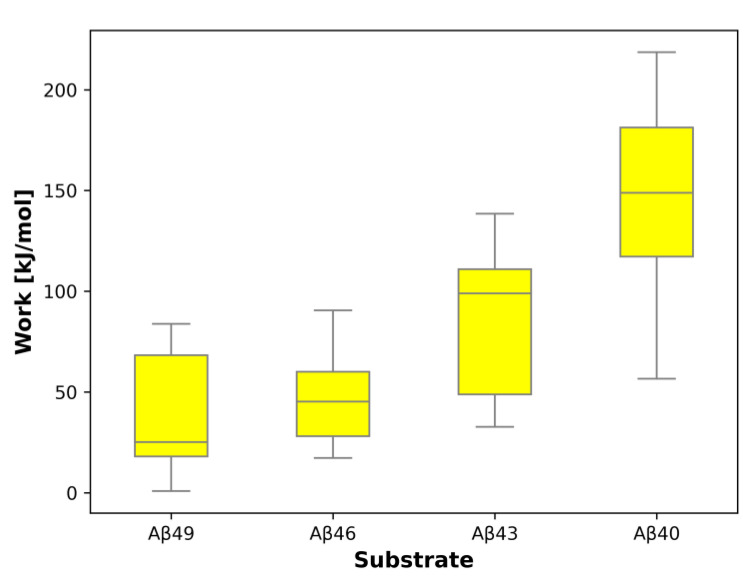
Box-plot representing the work/energy required to unfold the substrate by three residues in the active site of GS during SMD simulations. For each substrate, eight independent simulations were conducted. The work/energy was calculated for that frame with the shortest sum of distances of the n − 3 residue to the catalytic residues.

**Figure 10 ijms-25-02564-f010:**
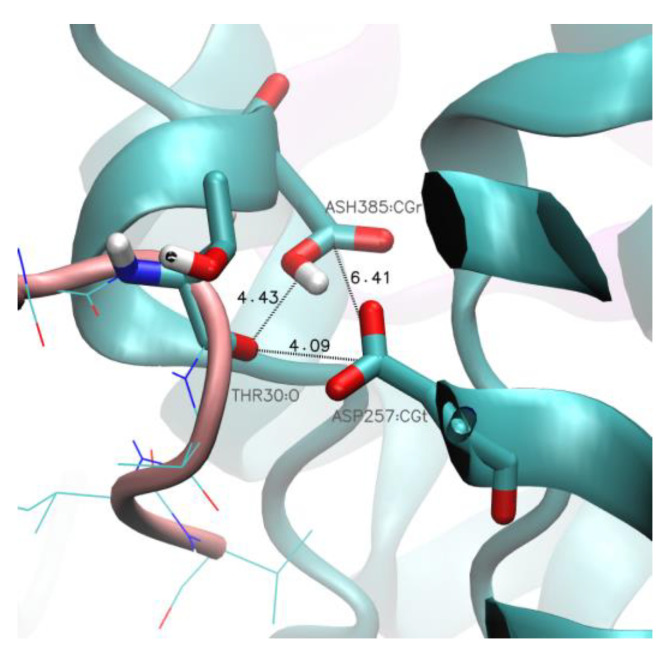
GS-SMD simulations—possible cleavage event. A frame with the shortest distance between GS catalytic residues (Asp257 and protonated Asp385) and the n − 3 substrate residue (Thr30) in the simulation of substrate unfolding Aβ_49_ → Aβ_46_. The sequence number of the C-terminal residue in the GS-SMD server is always 33, regardless of the thread sequence. Color scheme: PS-1 in cyan and Aβ in salmon.

**Table 1 ijms-25-02564-t001:** Mutations of residues Arg269, Arg377, and Lys380 and their biological effects.

Mutation	Biological Effect	Ref.
R269G	Early-onset AD; decreased protease activity with APP; increased Aβ_42_/Aβ_40_ ratio.	[30]
R269H	Early-onset AD; increased Aβ (42 + 43)/(37 + 38 + 40) ratio in cells; decreased GS activity.	[29,31,32]
R377M	Early-onset AD; uncertain significance, but in silico algorithm predicted it is deleterious.	[29]
R377W	In vitro, decreased Aβ_42_ production and abrogated Aβ_40_ production; nearly abolishes protease activity with APP.	[30,33,34]
K380R	Unknown, but in silico predictions suggest damaging effect.	[32]
Del377-381	Loss of Notch1 and APP-C83 cleavage.	[18,28]

## Data Availability

The data supporting reported results (MD simulations) are deposited at the Faculty of Chemistry, University of Warsaw, and are available upon request. The SMD simulations are available as shared jobs on the GS-SMD web server (https://gs-smd.biomodellab.eu).

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
