# Peer review of "Conformational Changes and Unfolding of β-Amyloid Substrates in the Active Site of γ-Secretase"

_ijms, 2024, doi:10.3390/ijms25052564_

Round 1

Reviewer 1 Report

Comments and Suggestions for Authors

The paper by Jakowiecki et al. presents significant findings, especially in identifying differences in the flexibility of the extended C-terminal part of β-amyloid substrates. These findings are contrasted between more and less amyloidogenic pathway substrates. The role of positively charged residues in the presenilin-1 component of γ-secretase, such as Lys380, Arg269, and Arg377, is highlighted for their potential facilitation in the stretching and unfolding of substrates, a crucial aspect of the cleavage process. The study also notes exceptionally high forces and work/energy required for pulling the Aβ40 substrate, which might explain its infrequent trimming in γ-secretase activity.

However, the study would benefit from further exploration in several areas:

1. Investigating how conformational changes influence the enzyme's activity or substrate specificity.

2. Assessing whether there is clinical mutational data or structural evidence supporting the roles of Lys380, Arg269, and Arg377 in γ-secretase.

3. Analyzing the pKa changes of Lys380, Arg269, and Arg377 during the MD simulation to understand how their charge states and interactions might influence γ-secretase's enzymatic activity.

4. Discussing the potential clinical implications and their contribution to the development of therapeutic strategies for Alzheimer's disease.

Overall, the manuscript is well-written and presents significant findings. Addressing these points would enhance the manuscript's contribution to our understanding of γ-secretase's role in Alzheimer's disease.

Author Response

The responses are in the file.

Reviewer 2 Report

Comments and Suggestions for Authors

Filepek and his colleagues conducted a study on the structural alterations of a group of β-amyloid substrates within the active site of the γ-secretase enzyme. The substrates mentioned are the primary constituents of amyloid plaques. The work utilized threading techniques to align the sequences of Aβ substrates with the most recent structure of γ-secretase in order to acquire an extended C-terminus of the substrates. The authors conducted all-atom molecular dynamics simulations to investigate conformational changes, and employed directed molecular dynamics simulations in an implicit membrane-water environment to expedite unfolding. The authors observe significant variations in the flexibility of the prolonged C-terminal region among substrates following the more and less amyloidogenic pathways. The study revealed that the calculated forces and work/energy required for pulling Aβ40 were remarkably high, which explains the infrequent occurrence of substrate trimming.

Throughout the content, the authors have successfully encapsulated the dynamic interplay between modern technologies and scientific innovation, underscoring its potential to revolutionize the field, making it more efficient and accessible. This article promises to offer valuable insights into the rapid developments in structural biology, emphasizing the importance of conformational changes of these molecules in disease progression.

Given the clarity, depth, and relevance of the content presented, I believe this article is well-positioned to contribute significantly to the current literature in the field. It will undoubtedly provide readers with a comprehensive understanding of the current state-of-the-art methods.

In general, the paper exhibits a commendable level of writing proficiency, and the examination of state of the art methods in studying amyloid peptides is intellectually stimulating. The authors' inclusion of fundamental concepts in initial section is deemed significant and contributes to the extent body of literature. Figures presented in the paper constitutes a significant inclusion that has the potential to enhance the paper's citation count.

 My first and only concern is the paper is not well referenced. I request the authors to cite recent literature that highly related to this study.

Author Response

The responses are in the file.

Reviewer 3 Report

Comments and Suggestions for Authors

Jakowiecki et al have investigated the conformational changes of β-amyloid substrates in the γ-secretase active site by molecular dynamics approach. I commend the authors for their thorough work. Employing molecular dynamics simulations and steered molecular dynamics simulations in an implicit membrane-water environment allowed for valuable insights into the flexibility and unfolding of substrates. The findings, particularly the differences in flexibility between more and less amyloidogenic pathway substrates and the role of positively charged PS-1 residues in substrate unfolding, are intriguing. Overall, a well-designed study with promising contributions to our understanding of AD. I have the following minor comments.

The full form of PS-1 should be provided in the abstract.

The line “These peptides are produced by the membranous γ-secretase complex through successive 12 cuts of the amyloid precursor protein cleavage product generated by β-secretase.” in abstract should be rephrased for clarity.

The color scheme in Figure 3 should be specified in legend. It is unclear what green and tan color proteins are.

Again, the color scheme in Figure 7e and 8c should be specified in legend. It is unclear what cyan and tan color proteins are.

 It seems some statistical analysis was used to derive the deviations in Fig. 10. The used statistical methods should be included in the methods.

I suggest the results should be discussed in more details for its implications for Alzheimer's disease pathology / therapy.

Author Response

The responses are in the file.

Round 2

Reviewer 1 Report

Comments and Suggestions for Authors

I am pleased with the authors' efforts in addressing the initial feedback. 

The authors have effectively implemented suggestions, notably developing the GS-SMD server for substrate specificity investigations, enriching the manuscript with mutational data, and discussing the implications for Alzheimer's disease therapeutic strategies.

Given the comprehensive revisions and the value of the findings, I am happy to recommend the manuscript for publication as it stands. The manuscript is a commendable contribution to the field and should be shared with the scientific community.

Reviewer 3 Report

Comments and Suggestions for Authors

All my queries are answered and manuscript can be accepted.